# Gender Perspective in Dual Diagnosis

**DOI:** 10.3390/brainsci11081101

**Published:** 2021-08-21

**Authors:** Marta Torrens-Melich, Teresa Orengo, Fernando Rodríguez de Fonseca, Isabel Almodóvar, Abel Baquero, Ana Benito

**Affiliations:** 1IMIM Hospital del Mar Medical Research Institute, Autònoma University, 08003 Barcelona, Spain; mtorrens@imim.es; 2El Grao Addictive Behaviours Unit, Clinical Hospital Valencia, 46024 Valencia, Spain; orengocaus.teresa@gmail.com; 3Institute of Biomedical Research of Malaga, 29010 Malaga, Spain; fernando.rodriguez@ibima.eu; 4Department of Psychobiology, Complutense University of Madrid, 28040 Madrid, Spain; 5Nursing Department, Cardenal-Herrera-CEU University, 12006 Castellón, Spain; isabel.almodovar@uchceu.es; 6TXP Investigation Group, Department of Medicine and Surgery, Cardenal Herrera-CEU University, 12006 Castellón, Spain; abel.baquero@uchceu.es; 7Proyecto Amigó-Proyecto Hombre Foundation, 12005 Castellón, Spain; 8Torrente Mental Health Unit, General University Hospital, 46900 Valencia, Spain

**Keywords:** gender perspective, dual diagnosis, women

## Abstract

Little data are available for women diagnosed with a dual diagnosis. However, dual diagnosis in women presents increased stigma, social penalties, and barriers to access to treatment than it does for men. Indeed, it increases the probability of suffering physical or sexual abuse, violent victimization, gender-based violence, unemployment, social exclusion, social-role problems, and physical and psychiatric comorbidities. Thus, a transversal sex and gender-based perspective is required to adequately study and treat dual diagnosis. For this, sex and gender factors should be included in every scientific analysis; professionals should review their own prejudices and stereotypes and train themselves specifically from a gender perspective; administrations should design and provide specific treatment resources for women; and we could all contribute to a structural social transformation that goes beyond gender mandates and norms and reduces the risk of abuse and violence inflicted on women.

Little data are available for women diagnosed with a dual diagnosis [1,2], to the extent that we do not even have clear data regarding the prevalence of dual diagnosis in the general female population. Available data indicates that dual diagnosis in women presents increased stigma [3,4], social penalties [3], and barriers to access to treatment [4,5] than it does for men; as gendered assumptions about appropriate behaviour for women (for instance acting as a wife and/or mother), societal disapproval of women’s use of substances, and the risk of losing their relationships may prevent help-seeking [4]. Indeed, it increases the probability of suffering physical or sexual abuse [2,6,7], violent victimization [8], gender-based violence [9], unemployment [4,10], social exclusion [10], social-role problems such as fulfilling family and work obligations [4], and physical [2,4,11] and psychiatric comorbidities [4,12].

Basic, preclinical, and clinical research has shown the presence of biological differences between the sexes from the beginning of embryonic development and throughout the entire life cycle. This dimorphism affects health, protective or vulnerability factors, social and relational life, the search for treatment, and responses to therapeutic interventions. There is evidence for genetic differences in stress-related effects, known to often mediate or modulate sex differences in addiction-related behaviours [13]. Differences have been detected in brain areas involved in craving, addiction, and relapse: the cerebral cortex (females showing a larger extent of cortical neuropil and lower neuronal numbers), the medial amygdala (approximately 20% smaller in females), and the caudate putamen and hippocampus (larger in females than in males) [14].

Animal studies have revealed sex-dependent differences: females and males differ for motivation to obtain a specific drug, levels of drug intake, or the propensity to reinitiate drug-seeking behaviour following a period of abstinence [14]. The estrous cycle is key in differences in reward and craving for drug [14].

Adult women have more gray matter in the medial prefrontal cortex (important for regulating executive function), while males have more gray matter in the anterior cingulate cortex (involved in hedonic and impulsive activity), what could lead to sex differences in the cycle of substance use disorders, including maintenance and relapse [15]. Estradiol would exacerbate drug use by increasing reinforcing effects [16] and sex differences in stress circuitry may explain sex difference in risk for comorbid alcoholism and stress-related disorders [17]. In several substances, women take less time to progress to dependence than men, although it is not clear whether the menstrual cycle has a similar effect to the estrous cycle in increasing motivation to self-administer substances [14]. Regarding alcohol use disorder, sex differences have been found in tryptophan metabolism [18]. Density and regional distribution of µ-opioid receptors vary between the sexes and, in females, across the ovarian cycle, while women have a higher number of D2-like receptors in the frontal cortex than men [14]. Besides, sex differences in the effect of drugs of abuse might be due, at least in part, to differences in muscle mass and fat tissue distribution between women and men, as well as in the gastric emptying time, which undergoes significant changes during the menstrual cycle [14].

However, studies on dual diagnosis continue to be carried out mainly in male patients [14] or focus only on differences between the sexes. Except for worthy exceptions e.g.: [19,20], we have little knowledge of the specific characteristics and needs of women, which in turn contributes to the androcentric design of interventions, resources, and treatment services. For this reason, women with dual diagnosis are reluctant to attend services at which they feel judged or, for example, feel they might risk losing their children [3,4].

Furthermore, female gender roles can act to precipitate dual diagnosis. Being a woman increases the probability of traumatic experiences such as abuse [7] and gender-based violence [21], which can hinder the development of adaptive coping strategies and can produce biological changes that themselves constitute vulnerability factors for substance use and mental illness [22]. In turn, substance use and mental illness are risk factors for further abuse and trauma [6,7], thus perpetuating the cycle of victimization. In parallel, domestic violence in childhood increases the risk of abuse in future relationships [9], maybe shaping girls into adopting behaviours typical of traditional female roles, which can foster emotional and economic dependence. This dependence also increases the risk of being initiated or induced into substance use by a partner [23], as women may abuse substances in an attempt to build or maintain relationships [4]. Moreover, a lack of adaptive coping skills (or the inability to put them into practice) is related to a reason for substance consumption in women: the desire to reduce emotional distress [20]. Therefore, women are more likely to self-medicate or use substances to deal with stress or pain [4]. Women are more likely to drink to regulate negative affect and stress reactivity, while men may be more likely to drink for positive reinforcement [15].

Thus, a transversal sex and gender-based perspective is required to adequately study and treat dual diagnosis. For this, sex and gender factors (e.g., caregiver role and unpaid work) should be included in every scientific analysis; professionals should review their own prejudices and stereotypes and train themselves specifically from a gender perspective; administrations should design and provide specific treatment resources for women; and we could all contribute to a structural social transformation that goes beyond gender mandates and norms and reduces the risk of abuse and violence inflicted on women.

National Institutes of Health required the inclusion of women in clinical research in 1993 and sex as a biological variable in basic and preclinical research in 2016 [15]. Research should distinguish biological sex from gender [24]; identify the precise role of hormones; and go deeper into the differential effects and health consequences that abused drugs may induce in women and men, the gender specific factors triggering drug use, the medical problem of drug use in pregnant women, and which environmental and sociocultural risk factors may contribute to drug abuse and relapse in women and men [14]. Studies should include careful task design, multiple levels of analysis and appropriate statistical modelling of sex considering a priori study hypotheses [25].

The diversity of user experiences (based on sex, gender, ethnicity, socio-economic level, ability, age, sexual orientation, etc.) should be recognized [4] and cared for. Women with dual diagnosis ask us to consider them, listen to their voices, and to create and adapt resources according to their specific needs and characteristics. Resources with conciliation, social integration, and autonomy promotion measures; and without language or sexist content are needed [3]. Treatment should broach gender stereotypes, acknowledge discrimination, and advocate for equality [4]. Comprehensive treatment devices for mental illness, addiction, and gender violence; with groups only for women [12]. Treatment related to family and trauma issues, with strategies specifically focused on reducing risk of abuse and coping with trauma [2]. Agencies should provide concrete assistance in prenatal health care, parenting education, and childcare; and residential programs should offer live-in options for children [4]. Programs incorporating childcare, parenting classes, job skills or employment enhancement, and specialized mental health treatment for trauma and comorbid mental illness [4]. They want services that meet their real needs, that stop blaming them, and that show them trust and the ability to listen, understand, and recognise their issues. Only in this way will women be able to sufficiently rebuild their identities and self-esteem and overcome the triple stigma of being female, mentally ill, and addicted.

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
