# Peer review of "Gender Perspective in Dual Diagnosis"

_brainsci, 2021, doi:10.3390/brainsci11081101_

Round 1

Reviewer 1 Report

I think this subject is important and I support  the message, but I have a problem with the fact that the authors provide so little scientific support for their thesis and formulate it one-sidedly.

For example, according to research dual diagnosis often leads to more victimisation, even in men. (see attachment) 

The literature cited is limited and recent research seems to be unknown.

The message would be more persuasive if the formulation were more nuanced, objective, and better substantiated.

Author Response

  • We have expanded the bibliographic references supporting our thesis.
  • We have included the reference de Waal 2018 that speaks of that dual diagnosis often leads to more victimization, even in men. In this article, when multiple logistic regression analysis for violent victimization is carried out, one of the factors that increases the risk is female gender. Namely, although both sexes are victimized, women are more so. This result is particularly significant given that women are less than 30% of the sample.
  • We have expanded the cited literature with recent research. However, in line with what is stated in our article, there is a lack of research in this regard. For example, searching for the MESH terms Dual Diagnosis and Sex Characteristics returns only 33 results, 4 in the last 10 years and 0 in the last 5 years.
  • We have tried to formulate the message more objectively, changing emotional adjectives to fewer emphatic words, and supporting it better with more data and bibliographic references.

Thank you very much for your suggestions, we believe that the article has been substantially improved.

Reviewer 2 Report

Dear authors,

I read this paper with great interest, but also with care. I (hereunder) provide some comments for the authors to consider in order to improve their paper:

I would advise to refrain from emotional and little nuanced adjectives such as “extraordinally little”, “we must all”, “we do know that”,..

Could the authors provide some additional referencing to international (preferably in the English language) literature regarding the statement that “What we do know is that dual diagnosis in women presents increased stigma, social penalties, and barriers to access to treatment than it does for men”? This would absolutely strengthen this statement.

I would very much like to see referencing on the statement “Basic, preclinical, and clinical research has shown the presence of biological differences between the sexes from the beginning of embryonic development and throughout the entire life cycle.” Do they mean in terms of brain functioning and neurostructure/fysiology (I would suppose since we are talking about dual diagnosis and psychiatric comorbidities)? However, to my knowledge there is also evidence that (at least in neuroscience), the gender differences may have been overrated, so this is often an interesting discussion. I would advice that the authors work this sentence out, and add more details in these differences they believe play a major role in dual diagnosis, as well as a hypothesis as how this influences dual diagnoses. If the statement of this paper is that this is crucial and we must all take gender in consideration at any analysis, this is in my eyes to be better worked out.

The authors state “They want services that meet their real needs, that stop blaming them, and that show them trust and the ability to listen, understand, and recognise their issues.” I do believe that this statement is to be considered for both male and female and consists of proper care. If a facility does not trust, listen, understand, and recognize patients’ issues (whether they are male of female), the problem lies in the treatment and not in the gender being treated. Perhaps authors could add that it is in the responsibility of the treatment centers and organizations to provide adequate care, irregardless of gender, with a specific focus on the additional needs of women with dual diagnosis. Perhaps the authors could add more specificity to what you would advise to implement to reach that goal and what this would specifically mean for women care.

Author Response

  • We have changed emotional adjectives for less emphatic words.
  • We have provided additional referencing to international literature in English language.
  • We have expanded the data regarding sexual and gender differences. However, in line with what is stated in our article, there is a lack of research in this regard. For example, searching for the MESH terms Dual Diagnosis and Sex Characteristics returns only 33 results, 4 in the last 10 years and 0 in the last 5 years.

         The discussion on whether sexual and gender differences have been                     overrated or ignored is very interesting, but we think it goes beyond the             objectives of this work and it will be better dealt with when there is more             research in this field.

  • We have added the phrase “The diversity of user experiences (based on sex, gender, ethnicity, socio-economic level, ability, age, sexual orientation, etc.) should be recognized [Brigth 2011] and cared for”. We have also expanded on suggestions to improve care for women with dual diagnosis.

Thank you very much for your suggestions, we believe that the article has been substantially improved.

Round 2

Reviewer 1 Report

I can agree to this version which has been updated meaningfully.

This manuscript is a resubmission of an earlier submission. The following is a list of the peer review reports and author responses from that submission.

Round 1

Reviewer 1 Report

The present perspective paper sough to identify the dearth of literature exploring the etiology, maintenance and treatment of concurrent disorders among women. While this topic is an area of much needed growth in the scientific literature and would likely be of interest to the readers of Brain Sciences, there are several shortcomings that prevent me from recommending it’s publication.

A large portion of this paper focus on the psychological impact of dual diagnoses among women (e.g., stigma, gender roles); however, the paper does not adequately address the emerging literature detailing the sex differences in underlying neurobiological mechanisms in dual diagnoses nor other important differences (e.g., stress-response systems). Many of the claims are made without citations (e.g., page 2,  lines 50-54). Additionally the authors do not make significant suggestions for improving the inclusion of women in research/clinical experiences. For instance, what are the identified specific needs for women with concurrent disorders? Are there special treatment considerations? Additionally, how do we improve the study of/inclusion of women in research (e.g., advocating for studies powered to explore sex differences?).

Reviewer 2 Report

Addressing the unique needs of women with co-occurring mental health and substance use disorders is an important topic.  This opinion piece needs to be framed within the larger body of literature already existing on this topic. For example, the following statement is not supported by the literature: “Women diagnosed with a dual diagnosis have been systematically ignored in research.”  In fact, women with dual diagnoses have been included in research studies (e.g., Essock SM, Mueser KT, Drake RE, Covell NH, McHugo GJ, Frisman LK, Kontos NJ, Jackson CT, Townsend F, Swain K. Comparison of ACT and standard case management for delivering integrated treatment for co-occurring disorders. Psychiatric Services. 2006 Feb;57(2):185-96.), some of which even speak to gender differences (e.g., Differences between men and women in dual-diagnosis treatment. American Journal on Addictions. 1997 Jan 1;6(4):311-7.; de Waal MM, Dekker JJ, Kikkert MJ, Kleinhesselink MD, Goudriaan AE. Gender differences in characteristics of physical and sexual victimization in patients with dual diagnosis: a cross-sectional study. BMC psychiatry. 2017 Dec;17(1):1-9.; Anne Comtois, K., & Ries, R. K. (1995). Sex differences in dually diagnosed severely mentally III clients in dual diagnosis outpatient treatment. American Journal on Addictions4(3), 245-253.). 

Similarly, in the second paragraph, the authors note that studies have not addressed unique needs of women which may lead to them to avoid services for fear of consequences, like losing custody of their children.  However, there are research studies that have addressed these issues (e.g., Morris SK, Schinke SP. Treatment needs and services for mothers with a dual diagnosis: Substance abuse and mental illness. Journal of Offender Counseling Services Rehabilitation. 1990 Apr 20;15(1):65-84.; Holdcraft LC, Comtois KA. Description of and preliminary data from a women's dual diagnosis community mental health program. Canadian Journal of Community Mental Health. 2009 May 12;21(2):91-109.).

There are also a number of statements that need citations (e.g., What we do know is that dual diagnosis in women is more serious and presents increased stigma, social penalties, and barriers to access to treatment than it does for men.; In turn, substance use and mental illness are risk factors for further abuse and trauma, thus perpetuating the cycle of victimization; In parallel, domestic violence tends to shape girls into adopting behaviours typical of traditional female roles, which can foster emotional dependence, further increasing the risk of abuse in future relationships.; On top of all this, we must not forget that a lack of adaptive coping skills (or the inability to put them into practice) is related to the main reason for substance consumption in women: the desire to reduce emotional distress.)  

Additionally, the citation listed does not support the following statement “Indeed, it increases the probability of suffering abuse, gender-based violence, unemployment, social exclusion, and physical and psychiatric comorbidities”.  The study cited, instead, noted that presence of dual disorders is associated with a lower quality of life but that this relationship was no longer significant when trauma symptoms were added to the model.  The authors note that presence of dual diagnosis in women may indicate a history of trauma and that support predicts quality of life.   Please either provide a different citation or modify the statement accordingly.